# Generative AI for Film Creation: A Survey of Recent Advances

## Abstract

*Generative AI (GenAI) is transforming filmmaking, equipping artists with tools like text-to-image and image-to-video diffusion, neural radiance fields, avatar generation, and 3D synthesis. This paper examines the adoption of these technologies in filmmaking, analyzing workflows from recent AI-driven films to understand how GenAI contributes to character creation, aesthetic styling, and narration. We explore key strategies for maintaining character consistency, achieving stylistic coherence, and ensuring motion continuity. Additionally, we highlight emerging trends such as the growing use of 3D generation and the integration of real footage with AI-generated elements.*

*Beyond technical advancements, we examine how GenAI is enabling new artistic expressions, from generating hard-to-shoot footage to dreamlike diffusion-based morphing effects, abstract visuals, and unworldly objects. We also gather artists' feedback on challenges and desired improvements, including consistency, controllability, fine-grained editing, and motion refinement. Our study provides insights into the evolving intersection of AI and filmmaking, offering a roadmap for researchers and artists navigating this rapidly expanding field.*

## 1. Introduction

In recent years, generative AI (GenAI) has made significant advances in video generation with diffusion models[18, 40, 70, 99, 101], 3D asset creation with Gaussian Splatting and NeRF-based models[6, 29, 35, 53, 56, 60, 100], and avatar synthesis[44, 85]. AI-driven content creation is becoming increasingly powerful, enabling AI filmmaking. Over the past few years, we have witnessed a growing number of AI-generated films.

However, the artistic and academic communities remain largely disconnected. Artists often lack insight into where stochasticity originates, why maintaining character consistency is so difficult, why descriptions of multiple characters in the same frame can lead to confusion, and why generating drone views is more challenging than close-up shots.

Conversely, researchers have little knowledge of effective artistic workflows and real-world creative needs—do artists truly require one-minute-long generated clips? What aesthetic standards do they seek? To what extent do they need controllability over character and camera movement? Which features should be prioritized for development?

In this paper, we analyze user survey data from the MIT AI Film Hack, a filmmaking hackathon that has collected hundreds of AI films over three years (2023, 2024, 2025). We examine the adoption rates of various GenAI tools based on the submission data. We also conduct a quantitative analysis of artists' concerns and expectations, and present case studies showcasing how artists efficiently utilize these tools in their creative workflows. We aim to provide a comprehensive overview of the AI filmmaking landscape, offering insights into current trends, best practices, and key challenges in AI filmmaking.

## 2. Related Work

### 2.1. Understanding Film Production and Video Generation

Traditionally, film production consists of three phases: pre-production (scriptwriting, storyboarding, character design), production (direction, cinematography and other departments working in tandem), and post-production (editing, special effects, sound design, and mixing) [21, 87–89]. With advances in multimodal AI, researchers have developed text-to-video models that generate coherent clips or full AI-driven shorts from minimal input, synthesizing both appearance and motion [40, 70, 99, 101]. In recent multimodal generative models research, certain pipelines demonstrate the ability to maintain internal continuity, style coherence, and believable character interplay across challenging scene transitions [25, 104, 111, 112]. Beyond 2D frames, AI extends into 3D and 4D asset generation, whese virtual actors integrate high-resohere models such as Neural Radiance Fields (NeRF), 3D Gaussian Splatting and dynamic 3D representations enable realistic scene synthesis and temporal consistency [6, 29, 35, 53, 60, 100]. These approaches allow filmmakers to generate and animate objects and environ-

ments over time, ensuring spatial and stylistic fidelity [92]. Another critical advancement is AI-driven avatar creation and human motion synthesis, where digital characters are synthesized using neural geometry and motion priors, producing full-body avatars with expressive facial features and physical plausible movement [33, 63, 77, 113]. These virtual actors integrate high-resolution textures and skeleton rigs to deliver nuanced performances, reducing the need for large-scale casting or motion capture [37, 38]. AI-driven avatars also offer style adaptability, allowing seamless transitions between photorealistic and artistic aesthetics without asset reconstruction [110]. By automating complex tasks across pre-production, production, and post-production, AI is reshaping the filmmaking pipeline, reducing costs, streamlining workflows, and enhancing creative control, ultimately enabling richer storytelling with fewer logistical constraints.

### 2.2. AI Film Workflow

The earliest AI–movie integrations primarily targeted perception tasks—scene analysis, object detection, or camera calibration—to partially automate editing and tagging [64–67, 93]. Today, AI spans all production phases to elevate both creativity and efficiency [19, 27, 46, 68, 69, 104]. In pre-production, generative language models help refine scripts and produce initial concept art [30, 47, 50, 55, 57]. Meanwhile, diffusion-based storyboard generation can block out potential shots—complete with lighting or basic character poses—by interpreting textual scene descriptions. During production, real-time vision algorithms perform automated camera positioning, track actors' locations, or match composite elements (like CGI props) to real set positions [20, 58, 75, 92]. Techniques can also incorporate generative volumetric backdrops, instantly turning minimal green-screen footage into richly detailed sets [34, 48, 53] . Finally, in the post-production phase, automated shot segmentation, character detection, and special-effects composition substantially reduce the labor intensity of traditional editing workflows [66, 75, 93, 115]. Meanwhile, large-scale datasets like MovieNet provide multi-modal annotations that establish standardized benchmarks for shot composition, narrative structure, and affective analysis [24, 74]. To enhance emotional impact, researchers focus on the three main media components—visuals [94, 104, 109], sound [79], and editing [5, 114]—and apply AI to music generation [11, 28, 95] and automated editing [86]. Moreover, classical narrative frameworks such as the Hero's Journey [90] and the Freytag's Pyramid [98] are now informed by deep learning models that offer automated analysis and generative support for script pacing and character development [55]. Overall, as generative models and multi-modal learning continue to advance, AI's role in filmmaking—spanning creation, stylistic coherence, and audience engagement—will likely deepen, creating new opportunities and challenges for the industry.

## 3. Survey

With the rapid advancement of generative AI (GenAI), many computer scientists and filmmakers are eager to understand how these tools are being integrated into film production. In this section, we examine the adoption rates of various GenAI technologies and how artists rate the impact of different factors on film quality. We also surveyed artists on their opinions of current GenAI tools and their expectations for future GenAI advancements in film production.

### 3.1. GenAI Tools Adoption Rate

Filmmaking traditionally involves scriptwriting, storyboarding, shooting, and post-production, including editing, music, and voiceover. With the emergence of generative AI, we aim to explore how AI can contribute to different stages of this pipeline. We surveyed the use of various GenAI tools in the MIT AI Film Hack[54, 105, 106], an event that challenges participants to create short films using AI. Running annually in 2023[105], 2024[106], and 2025[54], this film hack provides valuable insights into the utilization of AI in scriptwriting, video generation, 3D content creation, music composition, and voiceover (Table 1).

We observed that nearly all participants incorporated image or video generation in their film production. Despite the increasing realism of AI-generated visuals [59], most films retained a cartoonish style—likely because inconsistencies are less noticeable in animation than in realistic footage [54].

While video generation remains the primary AI-assisted tool, the MIT AI Film Hack organizers introduced a dedicated 3D generation track in 2024[106] to encourage broader adoption of AI-driven 3D content. This led to a notable increase in 3D tool usage from 0% in 2023 to 23.7% in 2025(Table 1). Although this growth is promising, adoption remains lower than video gen tools(Table 1), suggesting that AI-based 3D generation still faces challenges in meeting filmmakers' expectations.

By 2024, more than half of the films incorporated AI voiceovers (Table 1). Note that not all films include voiceovers, meaning the actual adoption rate of AI voiceover tools among films with voiceover is higher. Notably, non-native English speakers particularly embraced AI voiceover tools, using them to generate seamless, natural-sounding English narration for a global audience[76, 83, 84]. demonstrating how AI enables borderless artistic expression and accessibility.

AI-generated music and sound effects saw increasing adoption, rising from 12.5% in 2023 to over 50% in 2024 and 2025 (Table 1). Interestingly, in the 2025 competition, the winning music piece was created by a human composer [7], despite AI's significant presence. The blind judging process ensured that works were evaluated solely on quality of the music in helping emotion expression, suggesting that

| Category | 2023 (n=8) | 2024 (n=67) | 2025 (n=118) |
|---|---|---|---|
| LLM-assisted scriptwriting | 37.5% | - | 54.2% |
| AI-generated video | 87.5% | 95.5% | 100.0% |
| AI-generated 3D assets | 0.0% | 20.9% | 23.7% |
| AI-generated voiceovers | 0.0% | 59.7% | 53.4% |
| AI-generated music/sound effects | 12.5% | 50.7% | 54.2% |
| Blending real and AI-generated footage | 12.5% | 1.5% | 17.8% |

Table 1. Adoption rate of GenAI tools in the MIT AI Film Hack from 2023 to 2025. Percentages indicate the proportion of films utilizing each tool in a given year. A '-' signifies that the item was not surveyed in that year.

human-created music still holds an edge in capturing nuanced emotion and variation.

| Feature | Importance mean (s.e.) |
|---|---|
| Selecting appropriate video gen tools | 6.45 (0.073) |
| Crafting detailed prompts | 5.91 (0.117) |
| Providing detailed style descriptions | 5.65 (0.127) |
| Generating multiple iterations | 6.05 (0.116) |

Table 2. Best practices for using video generation tools surveyed from 110 artists. Importance rated on a scale of 0-7.

| Feature | Number of tools in one film mean (s.e.) |
|---|---|
| MIT AI Film Hack 2023 | 2.50 (0.327) |
| MIT AI Film Hack 2024 | 3.46 (0.210) |
| MIT AI Film Hack 2025 | 3.14 (0.136) |

Table 3. Number of video gen tools used in one film in the MIT AI Film Hack 2023,2024 and 2025.

| Feature | Importance mean(s.e.) |
|---|---|
| Choosing the right genAI product | 5.97 (0.118) |
| Writing detailed prompts | 5.33 (0.160) |
| Trying the generation multiple times | 5.34 (0.173) |

Table 4. Best practices for using 3D generation tools (n=65). Importance rated on a scale of 0-7.

## 3.2. GenAI Tools and Film Quality

Through surveys of experienced AI film creators, we gathered insights into user preferences on key aspects of video generation (Table 2) and 3D generation tools (Table 4).

People regarded selecting the right GenAI tools as the most critical factor in achieving high-quality films. The competitive landscape is evident in the diverse range of tools reported by users, including Midjourney, Kling, OpenArt, Runway, and Pixverse. Notably, artists in the 2025 MIT AI Film Hack used an average of three tools per film (Table 3), suggesting that different tools play complementary roles in meeting visual expectations, as no single tool fully replaces the others on the market.

Survey responses emphasized the need for multiple generation iterations to mitigate the impact of stochasticity in individual AI-generated outputs (Tables 2 and 4). Additionally, users highlighted the importance of crafting detailed prompts, often utilizing prompt rewriting tools, to achieve visually rich and appealing results in both 2D and 3D generation tasks(Tables 2 and 4).

## 3.3. Artists' Expectations for GenAI Tools

While user practices play a crucial role, the fundamental determinant of perceived video quality remains the underlying capabilities of GenAI models.

Analysis of artist surveys (Table 5) reveals that consistent character movement is the highest-priority feature for video generation tools, followed by camera control and overall character consistency. This aligns with artists' expectations for GenAI outputs to exhibit naturalness and coherence comparable to traditional filmmaking, as real-world footage inherently maintains spatial-temporal consistency due to physics.

Character consistency has always been a priority for users. In the first iteration of the MIT AI Film Hack in 2023, three out of eight films featured dogs as main characters—a strategic workaround for the limitations of AI-generated human characters[105]. Unlike humans, where small facial differences are easily noticeable, dogs tend to appear more similar with simple descriptions, making them a workaround for maintaining character consistency.

People express a strong desire for greater control over camera angles. In real-world filmmaking, camera movement follows a complex 3D trajectory, involving not just the degree of motion (e.g., "pan left 0.1–10") but also factors such as the starting position, focal length, and depth of field. Accurately describing these elements can be challenging. Additionally, even with precise descriptions, the model often struggles to generate certain perspectives, such as drone views or long-distance shots. This limitation is likely due to a lack of sufficient training data for these specific viewpoints.

We also observed a strong demand for controlling character movement (Table 5), as movement plays a crucial role in expression. Additionally, maintaining consistent movements is often essential for seamless transitions between adjacent clips, serving as a visual hook.

There is also significant interest in generating multiple characters within a single frame (Table 5), particularly among users aiming to create more complex narratives in longer video productions.

We also surveyed users' ratings of 3D generation tools and found lower satisfaction compared to video generation tools, suggesting that 3D GenAI still has significant room for improvement. Many users reported that the generated 3D meshes often lack the desired styles and proper mesh topology (Table 6). Additionally, many 3D generation tools struggle to create fine structures(Table 6), such as hollow designs or intricate details.

## 4. Case studies

This section presents a series of AI film case studies that explore the dynamic interplay between generative AI tools and filmmaking practices across six key domains: visual aesthetics, character creation, 3D generation, XR filmmaking, creative technology integration, and novel artistic expression. Drawing from winning films such as *CLOWN* [26], *For Pixi* [22], and *Metanoia* [41], these analyses dissect how filmmakers strategically leverage AI's strengths—such as rapid stylization, hybrid workflows, and abstract generation—while innovatively addressing its limitations, including temporal inconsistencies, rigid motion, and photorealism gaps.

### 4.1. Case study of visual

Visual storytelling in AI filmmaking requires unified aesthetic styling, coherent shot composition, and stable camera control. AI now offers robust tools to support these creative processes, expanding the boundaries of visual experimentation [18, 111].

#### Aesthetic Styling

Several AI approaches have emerged to create specific aesthetic styles and maintain consistency throughout a film, including prompt engineering, image references, LoRA model training, and Stable Diffusion's AnimateDiff [23][18]. Given the current limitations of AI, artists often supplement these techniques with hand-drawings, digital collaging, and traditional post-production to achieve desired effects.

First, artists could generate coherent styles with textual and visual references for AI tools. By referencing early multiflash photography studies of movement, *O.R.V. 8*[10] merged historical aesthetics with contemporary palettes. *A Dream About to Awaken* [81] leverages prompts derived from AI image interpretation tools applied to hand-drawn

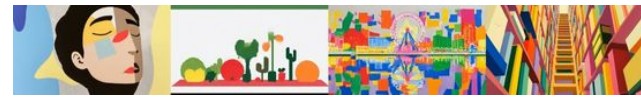

Figure 1. *A Dream About to Awaken* [81] Visual Style

storyboards, remixing them with diverse colors and styles to form a unique visual language.(Fig 1)

Second, hand sketches can serve as a stylistic foundation for generative AI. *Round Table* [62] began with hand sketching and digital collaging, using them as guides for AI-generated visuals to ensure consistency.

Third, training custom LoRA models [23] on curated datasets is another popular approach. In *Overthinking*[8], a specialized LoRA model [23] trained on 50 mid-century toy images evokes a nostalgic, minimalist style. This short film also incorporates Stable Diffusion's AnimateDiff [18], implemented through Comfy UI workflow[12] in conjunction with IPAdapter [36], to maintain stylistic consistency—particularly evident in elements like the chat bubbles.

Finally, traditional post-production techniques remain essential for polishing AI-generated visuals. *Qatsi* [7] integrated AI-generated abstract imagery with film grain and color grading, grounding its ethereal montages in the tactile texture of early cinema. *Round Table* [62] also manually refine and assemble AI-generated assets in Photoshop [2] and CapCut [42] to achieve a handcrafted, tactile feel.

#### Shot Composition and Camera Control

ControlNet guides Stable Diffusion effectively assists in shot composition[104] , and many AI tools already integrate camera control functionalities for creators to better control image and video generation[31] [40] . However, the duration of generated videos are typically under 20 seconds, making jump cuts necessary and posing a challenge to create a continuous viewing experience.

To address these limitations, artists have developed creative strategies. The "start and end frame" technique excels in camera movement processing and scene transitions, allowing for the display of content beyond the initial frame and enabling "one-take" effects. For example, *Invisible Women* [73] stitched AI-generated segments to produce seamless one-take sequences, creating a unique visual and narrative style. Also, utilizing 3D technology to control the camera is also an effective method. For instance, in *Dancestry* [16], AI was employed to assist in creating detailed 3D assets, movement rigs, and facial expressions. Animation and rendering were performed in Blender[9], allowing for precise control of camera movements.

#### Dataset Bias Challenges

Dataset limitations often introduce biases that pose challenges in visual generation. For instance, *Invisible Women* [73] highlights issues of gender bias, occupational algorithm

| Task | Importance | Current tools performance |
|---|---|---|
| Generate consistent characters according to the reference image/text description | 6.34 (0.103) | 4.45 (0.140) |
| Generate multiple main characters in one frame | 6.15 (0.100) | 3.91 (0.168) |
| Generate consistent character body movement | 6.62 (0.072) | 4.55 (0.147) |
| Follow the instruction in character body movement | 6.18 (0.103) | 4.06 (0.162) |
| Control camera movement | 6.35 (0.082) | 4.71 (0.126) |
| Allow local editing | 6.24 (0.102) | 4.33 (0.155) |

Table 5. Artists expectation for video gen tools for filmmaking(n=100)

| Task | Importance | Current tools performance |
|---|---|---|
| Generate decent meshes | 6.09 (0.127) | 3.99 (0.165) |
| Generate fine structures | 6.24 (0.115) | 3.84 (0.179) |
| Generate the desired styles | 6.28 (0.112) | 3.94 (0.166) |

Table 6. Artists expectation for 3D gen tools for filmmaking (n=65) Artists were asked to rate how well 3D gene tools perform in certain aspects and also how importance these features are

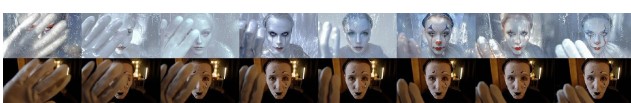

Figure 2. *Clown* [26] frame by frame style transfer

bias, and data disparities in AI-generated imagery processes. By optimizing their prompts and algorithm parameters in response to dataset limitations that tend to default to male representations, they ensure a more authentic portrayal of female characters and challenge entrenched stereotypes.

## 4.2. Case study of New Artistic Expression Forms with GenAI

While many AI filmmakers strive for visual consistency, some creators embrace the inherent limitations of generative AI to forge new artistic expressions.

### Embracing Randomness as a Creative Advantage

The film *CLOWN* [26] exemplifies how AI's unpredictable output can become a powerful narrative device. By employing a frame-by-frame stylization technique inspired by stop-motion animation, the creators transformed Midjourney's random generation feature into a psychological storytelling tool. [52]Each frame was individually processed through AI, maintaining visual continuity through a consistent art style while allowing subtle variations that mirror the protagonist's fragmented identity. This approach turned AI's inconsistency into a profound artistic statement about a clown gradually losing her sense of self. (Fig 2)

### Finding Poetry in the Imperfection of Image Generation

*Qatsi* [7] represents another approach. Rather than struggling against AI's limitations in photorealism, the creators embraced abstraction, the film embraced AI's abstract, expressionistic potential to evoke emotion and introspection.

The team adopted a monochromatic 4:3 aspect ratio aesthetic and abstract narrative following Soviet montage theory [15]. Inspired by the philosophy of filmmaker David Lynch [45] , the work embraces imperfections, irregularities, and errors—often seen as shortcomings—became tools for artistic expression rather than obstacles. By allowing technology to guide them toward poetic expression, they discovered unexpected beauty emerging from imperfection.

### Reimagining Traditional Animation Techniques with AI

Several projects revisited traditional stop motion animation methods through an AI lens. *Overthinking* [8] deliberately calibrated frame rates in After Effects to between 12-15 frames per second, both mitigating viewer discomfort from AI-generated motion artifacts and intentionally emulating stop-motion animation's distinctive aesthetic. Similarly, *Round Table* [62] merged AI-generated assets with traditional animation techniques, creating a unique workflow where AI-produced visuals were assembled in a stop-motion style, producing a handcrafted, tactile feel that countered the typically sleek appearance of AI imagery.

### Experimental 3D Aesthetics

Beyond these approaches, films like *Metanoia* [41] and *Dressage Marching Through Memories* [108] are beginning to explore advanced 3D technical innovations like Gaussian splatting—a technique that represents 3D scenes as a cloud of particles rather than polygonal meshes—opening new possibilities for visual representation through AI [29]. These experimental approaches suggest that as generative AI tools continue to evolve, the most compelling artistic expressions may come from creators who embrace AI's unique characteristics to develop entirely new visual languages.

Figure 3. *For Pixi* [22] Character Design and Consistency Control

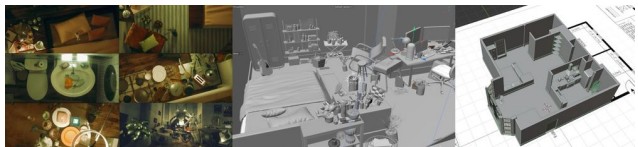

Figure 4. *Fish Tank* [102] final scenes, 3D Assets and model

### 4.3. Case Study of Character Creation with GenAI

Character creation is central to AI-assisted filmmaking, defining both the narrative and visual identities of AI-generated works. Among 118 film submissions in 2025, 109 featured characters, with 77 relying solely on digital characters, 16 blending real and digital elements, and 16 including only real actors.

**Character Design and Visual Consistency**

Films like *Tale of Lipu Village* [76] demonstrate how generative AI can merge disparate concepts—e.g., flowers shaped like fried eggs, sheep that grow broccoli—to yield imaginative character designs. Such unlikely combinations exploit AI's tendency to blend unrelated forms, resulting in cohesive aesthetics otherwise difficult to achieve via traditional methods. In *Dance of E-Spark* [97], the production team used style-specific prompts to create robotic characters, refining their lighting and poses in Photoshop. These examples underscore GenAI's capacity to produce distinctive visuals through informed prompting and supplemental manual edits.

Maintaining consistent character features across multiple scenes is especially challenging in AI-generated films. *For Pixi* [22] illustrated the importance of iterative refinement and prompt engineering by using the same prompt—"A claymation, puppet-style 3D animation world"—in every Midjourney [52] generation. Although subtle inconsistencies such as differing eye shapes persisted, tools like Midjourney's area-specific editor and 'vary' function proved invaluable. By generating 30–40 variations per prompt iteration and applying negative prompts to exclude unwanted elements, creators minimized undesirable discrepancies and sustained character uniformity throughout the film. (Fig 3)

**Character Motion and Emotion**

Generating lifelike motion remains one of the more complex areas of digital character creation. Approaches range from text-based prompts that describe movement to live-action motion capture. In *O.R.V. 8 Oscillating Rhythmic Vinyl* [10], a simple green screen setup allowed the performance of an actor captured on camera to be transformed into a fully animated robot using Wonder Studio [91]. The resulting sequences were composited into AI-generated environments

in Adobe After Effects [1] and Adobe Premiere Pro [3], providing a seamless mix of human expressiveness and AI-driven aesthetics. Believable emotional expression is vital for AI-created characters. *The Last Dance* [103]employed emotion-guided animation, where the film prompts specified how characters should feel and move in each scene.

**Character Voiceover**

AI-generated voices often struggle to convey emotional nuance and can sound mechanical. While some filmmakers supplemented AI dubbing with human voiceovers, others, like *Synthetic Rhythm*[82], ventured deeper into customizable voice-synthesis models. By carefully tuning pitch, tone, and pacing, and weaving these enhanced voices into the film's sound track, it created a captivating and authentic character voice.

In summary, AI-assisted character creation benefits from an array of strategies—prompt-driven design, meticulous consistency checks, hybrid motion capture, emotion-driven animation, and finely tuned voice models. By balancing AI's capacity for rapid, inventive output with manual refinement and human-led workflows, filmmakers can produce characters that are visually distinct, coherent across scenes, and emotionally compelling on screen.

### 4.4. Case Study of 3D Generation in Filmmaking

AI-assisted 3D modeling provides filmmakers with powerful tools for creating, integrating, and animating digital assets and scenes more efficiently than traditional 3D pipelines, as wekk as special effects.

**Mixing Real Footage and AI-Generated Content**

Seamlessly blending AI-generated 3D elements with live-action footage remains a central challenge. Films like *CLOWN* [26] employed Wonder Studio[91] , whose AI pipeline automatically animates, lights, and composes CG characters into real-world scenes. By adjusting lighting and depth, the film [91] ensures that digital character align with live-action camera angles and environments, preserving the authenticity of the hybrid sequences.

**3D Asset and Scene Generation**

*Fish Tank*[102], an experimental short, illustrates how AI drastically reduces the labor of 3D asset creation, but also reveals current limitations. Tools such as Luma AI [43], Meshy [51] and Hyper3d[13] cut asset generation time by more than

Figure 5. Volumography Scenes in *Former Garden*[96], *Metanoia* [41] and *Dressage Marching Through Memories* [108]

90%, producing a model in less than 5 minutes rather than 1–2 hours of meticulous sculpting. Luma AI[43]Captures real-world environments as high-fidelity and 3D meshes for scene creation. And Meshy [51]Quickly transforms prompts or 2D images into workable 3D models. In *Fish Tank*[102], the team used three distinct workflows: Photo-to-3D, MidJourney-generated Image-to-3D, and Text-to-3D, to expedite production and concentrate on storytelling.

Despite the advantages of AI-generated 3D assets, certain limitations persisted in the production of *Fish Tank*[102]. First, UV Mapping and Material Application: Meshy's [51]initial UV mapping capabilities were weak, requiring additional manual adjustments for accurate texturing. Later improvements, particularly quad-based topology, enhanced the usability of assets but still required refinement. Second, AI-generated mesh structures often lacked animation-ready topology. The rigid, mechanical nature of the generated models made them unsuitable for skeletal rigging and deformation without manual retopology. Third, AI struggled to generate complex, abstract objects with high creative flexibility, often defaulting to standardized geometric forms. This constrained its application in highly conceptual or surreal scenes. (Fig 4)

AI-based 3D generation continues to evolve, promising enhanced UV mapping, automated topology, and improved stylistic consistency. As these capabilities advance, AI will play an increasingly central role in lowering production costs, accelerating workflows, and empowering a broader range of creators to produce high-caliber 3D content.

### 4.5. Case Study of AI in XR Filmmaking

AI is rapidly reshaping Extended Reality (XR) filmmaking through two key advancements: AI-driven immersive videos and AI volumography.

#### AI-Assisted Immersive Videos

Immersive videos gain depth and interactivity from automated scene generation, motion capture, and real-time compositing. Tools like Wonder Studio[91] facilitate AI-driven motion capture by allowing creators to accurately capture intricate hand and body movements, which can then be blended into 360° footage for a more adaptive, responsive user experience. For instance, the *Machine Learning* [32] project uses AI-assisted compositing to integrate motion-captured animations directly onto 180° 3D video, creating a seamless XR environment. Project Reframe[71] was also used to capture hand movements for animation with headset cameras. The final film could be viewed in VR Headset.

#### AI Volumography

Volumography—powered by neural radiance field (NeRF) rendering, Gaussian splatting, photogrammetry, and volumetric video—goes beyond typical 2D or 3D video captures. Unlike traditional filmmaking's reliance on discrete frames, volumography records entire scenes as dynamic, navigable 3D/4D datasets. [29] [48] [53]

*Metanoia* [41], for example, captured a dancer using a NeRF-based pipeline and later explored a wide range of camera movements in post-production, reversing the usual order of shot planning. This non-linear process grants directors the freedom to experiment with pacing, composition, and viewpoint well after principal capture.

Similarly, *Former Garden*[96]demonstrates volumography's capacity for introspective storytelling, using a point cloud representation of fragmented memories to depict the protagonist's subconscious. (Fig 5)

By merging spatial computing with cinematic storytelling, volumography enables unlimited reshoots, unrestricted camera paths, and seamless real-CG integration in XR environments. Whether it's for immersive dance performances, urban documentation, or deeply personal narratives, these AI-driven approaches grant filmmakers new levels of flexibility and creative control. As a result, volumography is poised to become a cornerstone of future visual media, redefining how audiences engage with narrative spaces and making complex, interactive storytelling more accessible than ever.

### 4.6. Case Study of Creative Usage of Technology in Filmmaking

AI-driven filmmaking is not only about accelerating production pipelines but also about enabling novel, unconventional creative workflows that redefine how films are conceived and crafted. Filmmakers are pushing the boundaries of AI technologies to achieve unique artistic expressions.

#### Human-Machine Collaborative Approach

2023 Best Film Winner *DOG: Dream of Galaxy* [39] was created long prior to the emergence of advanced AI video generation tools. The project emphasizes a collaborative approach of human creativity with AI. The production began with script-driven prompt engineering to ensure narrative coherence. The Midjourney[52] generated images were processed through Stable Diffusion [78] to create depth maps. These depth-enhanced images were imported into Cinema 4D[49] , extruded into 2.5D models, allowing precise control over camera movement, focal depth, and composition. The film adopted a 4:3 aspect ratio, reminiscent of 1980s sci-fi films. Analog-inspired imperfections, such as film grain and scratches, were added in After Effects [1] to reinforce

| Pipeline Type | Typical Films | Typical Tools | Advantages | Challenges |
|---|---|---|---|---|
| 2D AI Pipeline (Text-Image-Video) | *For Pixi*[22], *Qatsi*[7],*Sacred Dance*[107] | OpenArt[59], Midjourney[52], Pixverse[61] | Quick iteration; strong visual style control; accessible tools | limited motion control; short video duration |
| 3D Generation Pipeline | *Dancestry*[16], *Overthinking*[8] | Meshy[51], Blender[9], Hyper3D[13], LumaAI[43] | Accurate modelling and camera control | Complex workflow; Time-intensive |
| Hybrid Live Action + AI | *Metanoia*[41], *Clown*[26], | Wonder Studio[91], Touchdesigner[14] | Emotive performance capture; Style remix; strong narrative flexibility | Integration complexity; Lighting and depth mismatch |
| XR / Volumetric Pipeline | *Former Garden*[96], *Metanoia*[41], *Machine Learning*[32] | NeRF[48][53], Unity3D[80], Unreal Engine[17] | immersive and interactive potential, post-capture camera control | High computational cost; dataset limitations |

Table 7. Overview of AI filmmaking pipelines, representative works, typical tools, benefits, and associated challenges.

the nostalgic aesthetic. AI-generated voices were enhanced with manual reverb effects, synchronized with electronic sound cues mimicking mechanical operations. This innovative workflow bridges AI-generated content with traditional filmmaking techniques, showcasing the potential of human-machine collaboration in cinematic expression.

**Hybrid Live Action, 2D and 3D Workflow**

*Metanoia* [41] reimagines AI's role in storytelling by merging live-action cinematography with generative AI tools. Set in a dystopian world where humanity is enslaved by digital masks, the film juxtaposes cold, algorithmic precision with the raw emotion of dance. The protagonist's struggle to reclaim her humanity unfolds through a hybrid workflow: Real-Time AI Integration: TouchDesigner [14] and Stable Diffusion [72] dynamically generate abstract visual textures, blending them with live-action footage to symbolize the encroachment of AI into her sanctuary. Luma AI's 3D scanning captured dancers' movements, which were then distorted using AI effects to create fluid, surreal transformations.[43] This fusion of physical performance and digital abstraction elevates dance as a metaphor for rebellion.

**Nonlinear Editing of AI-Generated Visuals and Music**

In *For Pixi* [22], nonlinear editing techniques were employed to synchronize AI-generated music and soundscapes with the evolving visuals. Rather than scoring music to a locked edit, the AI-generated music was produced iteratively alongside visual development, a synchronised back-and-forth between Premiere Pro [3]and Ableton Live[4]. This nonlinear interplay between sound and image led to a more cohesive, emotionally resonant audiovisual experience. The flexibility of AI-generated music allowed for continuous adjustments, enabling the filmmakers to refine the narrative rhythm dynamically during post-production.

These examples underscore how creative uses of AI technology extend beyond automation, opening new aesthetic possibilities and redefining artistic workflows in contemporary filmmaking.

## 5. Discussion

The case studies highlight both the vast potential and current limitations of AI-assisted filmmaking. While AI technologies have opened up unprecedented creative opportunities, ranging from character design to immersive volumetric storytelling, they also introduce challenges that require thoughtful human intervention. Based on the analysis across various domains including visual styling, character creation, 3D asset generation, motion, and sound, several key insights and recommended pipelines emerge for filmmakers seeking to adopt AI in their creative process. (Table 7)

## 6. Conclusion

Generative AI is reshaping filmmaking, significantly streamlining traditional workflows while simultaneously expanding the scope of creative storytelling. This survey reviewed recent advances and challenges in AI film creation, highlighting growing adoption in character animation, stylistic coherence, and immersive storytelling. While artists increasingly rely on generative tools for rapid asset generation and innovative visual styles, key challenges remain—particularly regarding consistency in character portrayal, nuanced motion control, and effective integration with real-world footage.

Looking ahead, continued progress in GenAI tools should prioritize improved controllability, finer-grained editing capabilities, and more intuitive camera management, aligning closely with filmmakers' practical and artistic needs. By addressing these critical areas, generative AI can further transform filmmaking into a deeply collaborative and accessible creative medium, paving the way for richer, more expressive cinematic narratives.

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
