# OpenReview forum: "Generative AI for Film Creation: A Survey of Recent Advances"
_thecvf.com/CVPR/2025/Workshop/CVEU — CVPR 2025_

### Official Review · Reviewer_j6vX · 2025-03-22

**Rating:** 2
**Confidence:** 4

**Review:**

This paper presents a summary/survey of generative AI for film creation. It analyzes survey data from MIT AI Film Hack and presents current workflows in the area.

Strengths:
- Paper writing: The paper is well written, well structured, and has many references that go beyond just presenting the survey data from the MIT AI Film Hack.
- Important topic: Generative AI for film creation is a hot topic, especially with emerging video models and their open access for artists.

Weaknesses:
- Missing diagrams: It would have been great to create diagrams summarizing specific workflows instead of just using text and sections. This would have been a nice contribution and a good summary of current workflows.
- Missing visuals: Moreover, there are basically no visual examples accompanying the text, making it more difficult to follow which part of the workflow is currently presented.

This paper is rather untypical, even for a workshop. I feel like the missing diagrams are for me a key reason to reject the paper. Currently, the contribution is very low and summarizing workflows and visuals into Figures would have been a nice contribution and could actually have decent impact.

---

### Official Review · Reviewer_Z88i · 2025-03-23

**Rating:** 4
**Confidence:** 4

**Review:**

Paper Summary:This paper provides a comprehensive survey of the application of Generative AI (GenAI) in filmmaking. It examines how GenAI tools like text-to-image diffusion, neural radiance fields, and avatar generation are transforming film creation through analysis of recent AI-driven films, and highlights GenAI’s contributions to character creation, aesthetic styling, and narration while exploring emerging trends such as the integration of real footage with AI-generated elements.

Paper Strengths:
1. Comprehensive Coverage: The paper offers a holistic view of GenAI applications across all stages of filmmaking.
2. It provides detailed case studies of award-winning films, demonstrating practical applications of GenAI in diverse filmmaking contexts.
3. The paper identifies emerging trends and future directions, such as advancements in 3D generation and XR filmmaking.

Major Weaknesses:
1. The paper lists LLMs in Table 1 but lacks detailed discussion on their integration with diffusion models for filmmaking. Tools like WeGen[1] and VideoAuteur[2] show potential in this area and could be highlighted.
2. As mentioned in Section 3.3, character consistency is crucial. Given that current mainstream video generation methods typically produce short clips of just a few dozen frames, it would be interesting to know how filmmakers maintain consistent character identities across these segments. The paper could provide some general ideas on this issue.

[1] WeGen: A Unified Model for Interactive Multimodal Generation as We Chat.

[2] VideoAuteur: Towards Long Narrative Video Generation.

---

### Official Review · Reviewer_ZzJq · 2025-03-25
**Survey paper for utilization of modern AIGC techniques in film making**

**Rating:** 4
**Confidence:** 4

**Review:**

This paper provides a broad survey of emerging Generative AI (GenAI) techniques for film creation, focusing on their adoption in independent projects and AI-centric film hackathons. It covers a wide range of areas including text-to-video generation, avatar synthesis, 3D content creation, hybrid live-action/AI workflows, and volumetric video capture techniques. The authors present a set of user survey results and case studies from the MIT AI Film Hack competitions (2023–2025) and discuss best practices, challenges, and feature requests from creative professionals. Overall, the paper highlights how GenAI is reshaping traditional filmmaking pipelines and identifies core design imperatives such as consistency, controllability, local editing, and multi-character coherence.

**Strengths**

- The paper offers a broad coverage of key domains (2D generation, 3D generation, avatars, volumetric capture) and highlights state-of-the-art approaches with plenty of real-world examples. This breadth provides a useful one-stop reference for those interested in how GenAI fits into various stages of filmmaking.
- Practical Insights from Filmmakers: Gathering feedback from the MIT AI Film Hack participants adds strong practical value. The paper goes beyond purely technical metrics and delves into actual user concerns (e.g., character consistency, camera control, system integration), which is highly beneficial for bridging the gap between research and practice.
- The inclusion of detailed mini “production diaries” (e.g., DOG: Dream of Galaxy, For Pixi, Fish Tank, Metanoia) is particularly helpful. These illustrate the workflows that teams used to mix multiple AI tools, manually refine outputs, and address known model limitations.
- AI-driven filmmaking is a rapidly emerging trend. This paper’s timely focus on practical usage, limitations, and future directions is valuable for both the research community and creative industry professionals.

**Weaknesses**
- The findings strongly emphasize that “maintaining character consistency” and “camera trajectory control” are top priorities. A deeper discussion on potential technical solutions—like identity embeddings, reference layers, or specific consistent-depth workflows—would be beneficial to the academic audience.
- The conclusion touches on future improvements (e.g., finer-grained editing, better 3D generation). It might be beneficial to incorporate a structured set of research directions that directly align with the user-requested features highlighted in Table 5 and Table 6, possibly with references to relevant cutting-edge techniques.

Overall, while this paper is not really technical, it stands out for its practical orientation and suitability for the theme of this workshop, bridging academic and artistic perspectives on AI-driven filmmaking. Despite some methodological and technical gaps, it is a strong workshop contribution that should stimulate further discussion among researchers and practitioners working on generative models for media production. Accepting it would enrich the workshop with its focus on real creative pipelines, user needs, and the nuanced interplay between automated generation and manual artistry.

---

### Decision · Program_Chairs · 2025-03-25

**Decision:**

Accept

**Comment:**

The paper provides a comprehensive survey of Generative AI applications in filmmaking, offering valuable insights from practical experiences and detailed case studies from the MIT AI Film Hackathons. Reviewers appreciated the broad coverage of methods, practical orientation, and timely focus on bridging academic research with creative workflows. However, concerns were raised regarding the lack of detailed visual diagrams, limited technical depth on maintaining character consistency, and insufficient integration of recent developments such as LLMs.

Given the overall positive feedback and relevance to the workshop theme, the paper is accepted. Authors are encouraged to enhance the camera-ready version by adding workflow diagrams, visual examples, and deeper discussions of technical solutions aligned with user-identified challenges.